# Assessing the genetic diversity of Ethiopian indigenous goat ecotypes at the hemoglobin locus and its associations with morphometric traits

Kebede Tilahun[1,2]*, Aberra Melesse[1], Simret Betsha[1]

**1** School of Animal and Range Sciences, Hawassa University, Hawassa, Ethiopia, **2** Department of Animal Science, College of Dry land Agriculture, Kebridehar University, Kebridehar, Ethiopia

* keedetilahun2000@gmail.com

## Abstract

Genetic variation is the baseline for viability, fitness, survivability, and improvement of livestock raised under diverse agroclimatic conditions. The study aimed to assess the genetic variation in the population of goats at the hemoglobin (Hb) locus and its association with morphometric traits. The blood samples were collected from 225 mature goats of both sexes. Morphometric traits, including body weight, chest girth, chest depth, body length, rump length, rump height, rump width, height at withers, fore canon circumference, and ear length, were measured to examine associations with the Hb genotype. The red blood cells were subjected to gel electrophoresis to determine hemoglobin variants. Hemoglobin polymorphisms were analyzed using the Pop gene (ver.32) and the associations with morphometric traits were analyzed using the statistical analysis system. Three Hb genotypes, Hb$^{AA}$, Hb$^{AB}$, and Hb$^{BB}$ were detected with genotypic frequencies of 0.34, 0.47, and 0.19, respectively. Hb$^{AA}$ was most frequent in highland goats (47%), while Hb$^{AB}$ was the predominant genotype in lowland and midland populations. A significant deviation ($p < 0.05$) from the Hardy-Weinberg Equilibrium (HWE) was observed for the midland goats, while those in the lowland and highland were under HWE. The heterozygosity ranged from 0.36-0.61, indicating population variability and potential for genetic improvement. Hemoglobin showed significant ($p < 0.05$) influence on most of the morphometric traits in which goats with the homozygous Hb$^{BB}$ genotype exhibited superior performance compared to other genotypes. In conclusion, hemoglobin polymorphism could be a viable option for assessing genetic diversity and may serve as a potential genetic marker to support selective breeding programs.

**Data availability statement:** All relevant data are within the manuscript and its Supporting Information files.

**Funding:** The research was funded by Hawassa University, Office of Vice President for Research and Technology Transfer (VPRTT) (Grant code- 445564844). The funders had no role in the study design, data collection and analysis, publication decision, or manuscript preparation.

**Competing interests:** The authors have declared that no competing interests exist.

## Introduction

Assessing the genetic diversity and current status of animal genetic resources is essential for sustainable utilization, improvement and conservation [1,2]. Genetic diversity enables the monitoring of the gene flow in populations, measuring the level of inbreeding [3] and evaluating environmental adaptation [4,5]. It is also the baseline for survival and productivity [6,7] and important to evaluate the magnitude of genetic improvement achievable in a population [8,9]. The variability of breeds and populations are important due to existence of variations in climate, nutrition, management, diseases causing agents, and socio-cultural status of the community. Local breeds are an invaluable source of genetic diversity [10] and assessing this variability is vital for selection and development of new breeds in response of change in the demand, climate, market and breeding goals [11,12].

Genetic variation can be evaluated based on morphological, physiological, productive performances and behavioral features. However, such character underestimates the level of genetic variation [13,14]. Therefore, the use of polymorphic variants of proteins, enzymes, and blood groups could provide an accurate estimate of genetic difference and prove their reliability as a genetic marker for selection of some traits of economic interest [15,16]. The polymorphic markers, which associated with traits, are helpful to select the right genotype at early age [17]. Hemoglobin is essential for examining adaptive change because it closely connects metabolic activities with external conditions [18]. Due to its biochemical, biophysical and physiological properties, hemoglobin and its polymorphic variants are associated with morphological, performance and adaptation traits [19] and have great importance in selection of animals for improved productivity. Hemoglobin (Hb) polymorphisms play a significant role in the study of population relationships [20], environmental adaptability and fitness [14,21]. For instance, sheep with the Hb$^{AA}$ genotype have been reported to exhibit greater tolerance to helminth infections than those with the Hb$^{BB}$ genotype [22]. The Hb$^A$ allele offers a selective advantage, particularly in high-altitude environments, due to its high oxygen affinity which supports better survival under hypoxic conditions [23,24]. Conversely, the Hb$^B$ allele has a lower hematocrit value, reduced blood viscosity and higher water content, traits that are advantageous to adapt and survive in arid lowland areas [6]. Beyond environmental adaptability, Hb variants have also been linked to animal productivity [25]. They have shown associations with productivity and fertility [22,26], hair length [27], wool quality [28], growth performance traits [29], morphometric traits and hematological profiles [24,30,31], survival rate of the offspring [32], milk yield [33] and reproductive performance [34].

Hemoglobin variants are related to body measurement traits in different livestock species. In goats, for instance, the Hb$^{AA}$ genotype exhibited higher body weight and chest girth [24], while the Hb$^{AB}$ has been associated with larger scrotum circumference, high milk yield, and body size [33,34]. Similarly in cattle, individuals with the Hb$^{AB}$ showed higher performance in most traits [35]. The observed associations between Hb types with morphometric traits suggest that hemoglobin polymorphisms [17,22] can serve as a potential biochemical marker and help in the selection of superior animals for genetic improvement programs [21,27,36]. Moreover, hemoglobin is

essential for measuring genetic variation with higher accuracy [37]. In areas where access to advanced molecular technologies is limited, Hb typing when integrated with phenotypic data [38,39] can provide a practical and cost-effective tool for detecting genetic diversity and guiding breeding decisions [40,41]. Despite, its relevance in detecting genetic variation and informing selection programs, hemoglobin polymorphisms of Ethiopian goats particularly those in the southern regions remains largely unexplored. To the author's knowledge except the study of [42], who examined the protein polymorphisms of four indigenous goat populations (Afar, Hararghe Highland, Western Highland and Western Lowlands), no research has been done to evaluate the genetic variations of goats reared across diverse agroecological zones of southern Ethiopia. Furthermore, that study did not consider the effect of agroecological zones or explore potential associations between hemoglobin variants and performance traits. Therefore, the objective of the present study was to quantify the magnitude of genetic variation and relationships at the hemoglobin locus and examine the associations between hemoglobin variants and morphometric traits in indigenous goat populations.

## Materials and methods

### Ethics approval

The Hawassa University Research Ethics Review Committee (RERC) examined and approved the study's experimental design and sampling procedures (Reference No: REC 016/23).

### Study area

The study was conducted in southeastern Ethiopia, in the Sidama and Oromia regions. Districts were purposely selected based on their goat population potential and agro climatic conditions. Three districts representing the three agroecological zones (highland, lowland, and midland) were selected. Of these, the two districts representing the lowland and midland agroecological zones were selected from the Sidama region while the highland agroecological zone was considered from the Oromia region. Loka Abaya is a district that represents lowland agroecological zone. It is situated at 6°42′-7°83′ North latitude and 37°92−39°14′ East longitude. Chuko is situated at 6°4 6'-7°01' North latitude and 38°04'-38°24' East longitude, which represents the midland agroecological zone. Kofele is the third district located at 06°50' to 07°09' North latitude and 38°38' to 39°04' East longitude and represents the highland agroecological zone.

### Experimental animals

A total of 225 adult goats (75 from each agroecological zone) of both sexes were sampled. The age of the goats was estimated based on dentition, and all sampled animals belonged to a similar age group, characterized by four pairs of permanent incisors (4PPI), indicating an age of four years and older [43]. To avoid the likelihood of sampling related animals, one to two unrelated individuals were selected per flock, and the farmers were consulted to verify the lack of relatedness of their animals. From each sampled goat, blood samples and morphometric data were collected. Morphometric traits including body weight, chest girth, chest depth, body length, rump length, rump height, rump width, height at withers, fore canon circumference, and ear length, were measured following FAO's guidelines [44]. Body weight was measured using a portable digital weighing balance having a 50 kg capacity with ±10g precision, while linear body measurements were taken using a centimeter-scale measuring tape. All measurements were taken early in the morning before the goats were released for feed and water to avoid the feeding effect on trait measurements.

### Experimental animal management system

The goat population in southeastern Ethiopia was commonly referred as Arsi-Bale type and distributed in Arsi, Bale, and Sidama [45]. They are dominantly reared under a free grazing system. However, there is a difference in production systems between agroecological zone in which tethering is practiced in the Sidama while herding is practiced in the highlands

of Arsi and Bale. The main grazing areas in the highlands were waterlogged valley bottoms and hillsides with supplementation of crop residues mainly barley and wheat straw [45]. In the Sidama region communal grazing land, cut grass and browsing on fallow land and crop residues are the main sources of feed for goats. Bore and rainy water are provided for goats in dry and rainy seasons, respectively [46].

### Consent with the participants

Before the data collection, households possessing at least three mature goats of both sexes were selected from each district, representing different agroecological zones. The purpose and nature of the data to be collected including blood and morphometric measurements, were clearly explained to the participating farmers. Awareness raising sessions were conducted by the researchers and extension workers in each *kebeles* (the smallest administrative unit in Ethiopia). Morphometric data were collected by researchers while blood samples were collected by experienced veterinarians.

### Blood sample collection

Four ml blood samples were collected from the jugular vein of each goat using a sterile syringe fitted with 18-gauge needles. The blood sample was immediately transferred into ethylene-diamine-tetra acetic acid (EDTA) coated tubes to prevent coagulation, and each test tube was properly labeled. The tube containing blood was then immediately placed in a box containing ice packs to maintain cold chain conditions and transported to the Animal Biotechnology Laboratory at Hawassa University for further processing.

### Hemoglobin electrophoresis

Red blood cells were separated from whole uncoagulated blood by centrifugation. Two (2 ml) of each blood sample was placed into a clean test tube, 4 ml of cold saline was added, and the obtained mixture was centrifuged at 4000 rpm for 10–15 min at 4°C [14]. The supernatant was discarded, and the sample was washed with cold saline. Then, the separated red blood cells were lysed by the addition of an equal volume of distilled water to expose the hemoglobin. The lysate was stored under refrigeration and subjected to electrophoresis. Hemoglobin genotypes were determined via 1% agarose gel electrophoresis (1g of agarose dissolved in 100mL of 0.5x TBE buffer). The solution was transferred to a gel cast tray with combs that formed wells. The gel was carefully transferred to the electrophoresis tank and filled with 0.5x TBE buffer (PH 8.4–8.5). A sample size of ten microliters was impregnated with a constant voltage of 120 volts and then run for 2hrs. After the completion of the electrophoretic run, the gel was carefully removed and inserted into gel documentation system that connected with computer and the hemoglobin pattern was visualized. The hemoglobin patterns were then read directly from the computer and identified based on their band movement. The fast band is designated Hb$^{AA}$, whereas the slow band is Hb$^{BB,}$ and the existence of two bands is designated Hb$^{AB}$.

### Statistical analysis

Hemoglobin genotype data were analyzed using Popgene32 software [47], to estimate allele frequency and genetic diversity parameters such as heterozygosity (Het), effective number of alleles (ne), and Nei's standard genetic distance. The observed genotype frequencies were calculated as follows:

$$\text{Genotype frequency of Hb}^{AA} = \frac{Number\ of\ Individuals\ with\ HbAA}{Total\ number\ of\ individual\ sampled} * 100$$

$$\text{Genotype frequency of Hb}^{AB} = \frac{Number\ of\ individuals\ with\ HbAB}{Total\ number\ of\ individuals\ sampled} * 100$$

$$\text{Genotype frequency of Hb}^{\text{BB}} \ = \ \frac{\textit{Number of individuals with HbBB}}{\textit{Total number of individual sampled}} * 100$$

The effects of agroecological zone and hemoglobin genotypes on morphometric traits were computed using the GLM procedure of the Statistical Analysis System [48]. Agroecological zone and hemoglobin genotypes were fitted as fixed effects. When the F test showed significant differences, least square means were compared via the Tukey–Kramer test. The raw data utilized for the analysis of morphometric characteristics and hemoglobin polymorphisms is accessible in S1 File of the Supporting Information. The statistical model used to test the influence of hemoglobin genotypes on morphometric traits was as follows:

$$Y_{ij} \ = \ \mu + A_i \ + H_j + \ e_{ij}$$

Where $Y_{ij}$= response of the observed variables (morphometric traits).

μ= overall mean.

$A_{i =}$ the effect due to $i^{th}$ agroecological zone $_,$ (i = highland, midland and lowland).

$H_j$ = the effect due to $j^{th}$ hemoglobin type (j = Hb$^{AA}$, Hb$^{AB}$ and Hb$^{BB}$).

$e_{ij}$ = random error.

## Results

### Effects of agroecological zones on morphometric traits

All the traits considered were significantly ($p < 0.05$) affected by agroecological zones (Table 1). Compared with those reared in the highlands, the lowland goat revealed significantly greater values for all morphometric traits except chest depth. Lowland goats had significantly higher ($p < 0.05$) body weight, chest girth, height at the withers, rump length, rump width, and ear length than midland and highland goats. In contrast, midland goats exhibited higher values for body weight, height at withers, body length, rump length, rump width, rump height, fore canon circumference and ear length compared to those raised in the highland agroecological zone (Table 1).

### Gene and genotype frequencies

The results revealed that the hemoglobin locus in the goat populations was polymorphic and presented three hemoglobin genotypes, namely, Hb$^{AA}$, Hb$^{AB}$ and Hb$^{BB,}$ with frequencies 34, 47, and 19%, respectively (Fig 1). The three genotypes are produced by two codominant alleles, Hb$^{A}$ and Hb$^{B}$. The observed genotype was varied among the goats in the three agroecological zones (Fig 2). In the highland goat population, the highest genotype frequency was observed for Hb$^{AA}$ (47%),

**Table 1. The effects of agroecological zones on some morphometric traits of goats.**

| Agroecological zone | BW | CG | HW | BL | RL | RW | RH | CD | FCC | EL |
|---|---|---|---|---|---|---|---|---|---|---|
| Midland | 29.0$^b$ | 72.1$^b$ | 67.8$^b$ | 64.2$^a$ | 20.7$^b$ | 14.8$^b$ | 67.5$^a$ | 31.0$^a$ | 8.0$^a$ | 13.8$^b$ |
| Highland | 26.3$^c$ | 71.1$^b$ | 62.8$^c$ | 65.7$^b$ | 19.5$^c$ | 13.7$^c$ | 63.7$^b$ | 29.7$^{ab}$ | 7.6$^b$ | 12.7$^c$ |
| Lowland | 31.5$^a$ | 75.4$^a$ | 69.6$^a$ | 65.8$^a$ | 21.5$^a$ | 15.4$^a$ | 68.3$^a$ | 30.3$^b$ | 8.3$^a$ | 14.9$^a$ |
| SEM | 0.609 | 0.669 | 0.478 | 0.474 | 0.203 | 0.184 | 0.459 | 0.238 | 0.081 | 0.126 |
| P-value | <.001 | <.001 | <.001 | <.001 | <.001 | <.001 | <.001 | 0.001 | <.001 | <.001 |

$^{a-c}$ = Means under the same column with different superscript letters are significantly different at $p < 0.05$; SEM = standard error of the mean; BW = body weight; CG = chest girth; HW = height at withers; BL = body length; RL = rump length; RW = rump width; RH = rump height; CD = chest depth; FCC = fore canon circumference; EL = ear length.

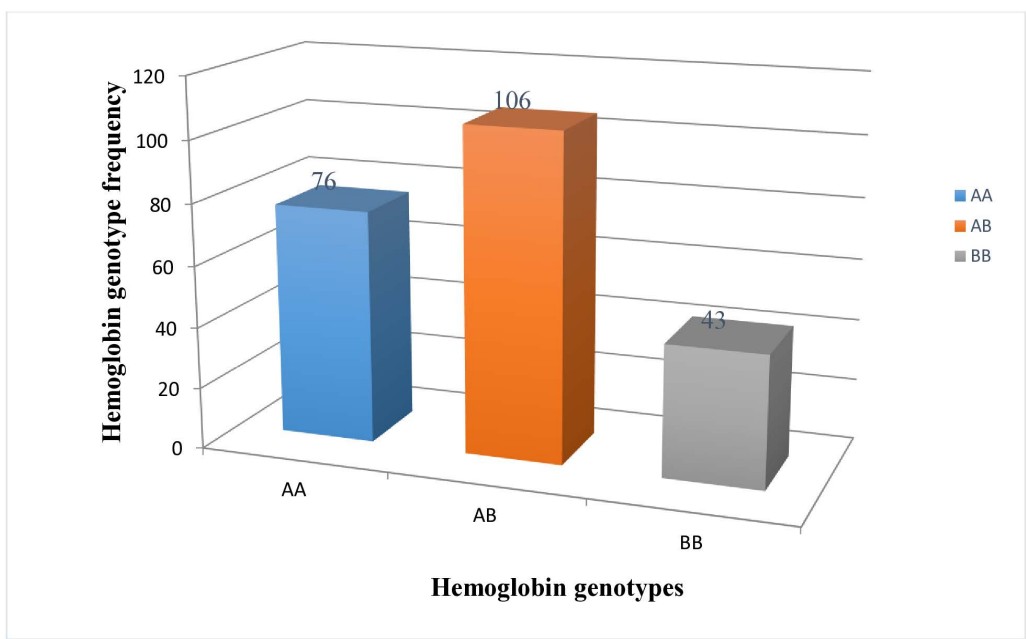

**Fig 1. Hemoglobin genotype frequency in goat populations.**

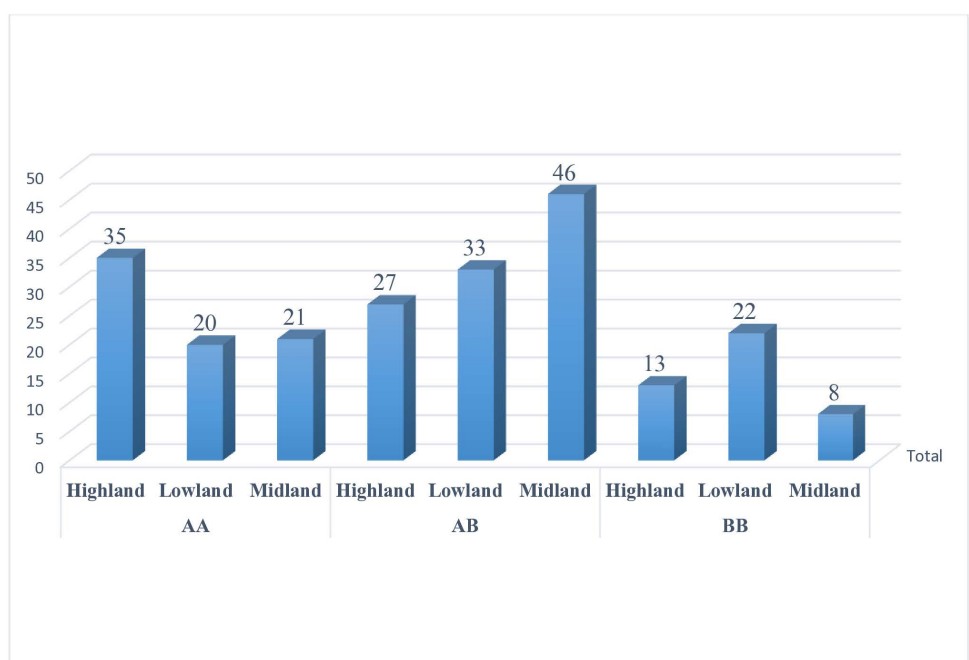

**Fig 2. Distribution of hemoglobin genotypes in the three agroecological zones.**

whereas, the highest proportions of $Hb^{AB}$ were 61% and 44% for the midland and lowland goat populations, respectively (Table 2). The frequency of allele A was higher 0.65 and 0.59 for the highland and midland goat populations, respectively. Allele B (0.51) was found to be abundant in lowland goats. The chi-square test was non-significant (p > 0.05) for highland and lowland agroecological zones, revealing that the goat populations are under HWE. However, it was significant (p < 0.05, $\chi^2 = 4.995$) for the midland goat population (Table 2), indicating that they are not under HWE. The distributions of Hb% for the $Hb^{AA}$, $Hb^{AB}$ and $Hb^{BB}$ genotypes in female goats were 36, 48, and 16%, respectively, whereas the corresponding values for male goats were 26, 46, and 28%, respectively (Table 2). The detailed results of hemoglobin polymorphisms for the three agroecological zones and the sex of goats can be found in the S2 and S3 Files of the Supporting Information.

## Genetic diversity

The observed heterozygosity (Ho) ranged from 0.36 for the highland to 0.61 for the midland goat populations (Table 3). The overall observed heterozygosity in the present goat population was 0.47. The mean expected heterozygosity (He) was 0.49. The effective number of alleles (ne) ranged from 1.84 to 1.99, with an overall average of 1.96.

## Nei's genetic identity and distance

The highest genetic similarity (0.99) was observed between the highland and midland goat populations (Table 4). On the other hand, the greatest genetic distance (0.05) was observed between the lowland and highland goat populations.

**Table 2. Gene and genotype frequency of indigenous goat populations based on agroecological zone and sex.**

| Populations | Genotype | Genotype Frequency | Genotypes | | Allele frequency | | $\chi^2$ |
| --- | --- | --- | --- | --- | --- | --- | --- |
| | | | observed | Expected | A | B | |
| Highland | | | | | 0.65 | 0.35 | 3.604[ns] |
| | AA | 0.47 | 35 | 31.2 | | | |
| | AB | 0.36 | 27 | 34.5 | | | |
| | BB | 0.17 | 13 | 9.25 | | | |
| Lowland | | | | | 0.49 | 0.51 | 1.192[ns] |
| | AA | 0.27 | 20 | 17.6 | | | |
| | AB | 0.44 | 33 | 37.7 | | | |
| | BB | 0.29 | 22 | 19.6 | | | |
| Midland | | | | | 0.59 | 0.41 | 4.995* |
| | AA | 0.28 | 21 | 25.7 | | | |
| | AB | 0.61 | 46 | 36.6 | | | |
| | BB | 0.11 | 8 | 12.7 | | | |
| Sex | | | | | | | |
| Female | | | | | 0.60 | 0.40 | 0.017[ns] |
| | AA | 0.36 | 61 | 60.6 | | | |
| | AB | 0.48 | 80 | 80.8 | | | |
| | BB | 0.16 | 27 | 26.6 | | | |
| Male | | | | | 0.49 | 0.51 | 0.528[ns] |
| | AA | 0.26 | 15 | 13.6 | | | |
| | AB | 0.46 | 26 | 28.7 | | | |
| | BB | 0.28 | 16 | 14.6 | | | |

ns = non-significant (p>0.05); *= significant at p<0.05.

**Table 3. Measure of genetic variability at the hemoglobin locus for the goat populations reared in three agroecological zones.**

| Agroecological zone | Sample size | Heterozygosity | | ne | Average heterozygosity |
| --- | --- | --- | --- | --- | --- |
| | | observed | expected | | |
| Highland | 75 | 0.36 | 0.46 | 1.84 | 0.48 |
| Lowland | 75 | 0.44 | 0.50 | 1.99 | 0.48 |
| Midland | 75 | 0.61 | 0.49 | 1.94 | 0.48 |
| Overall | 225 | 0.47 | 0.49 | 1.96 | 0.48 |

ne = effective number of alleles.

**Table 4. Nei's original measure of genetic identity (above diagonal) and genetic distance (below diagonal) between the goat populations reared in the three agroecological zones.**

| Agroecological zone | Highland | Lowland | Midland |
| --- | --- | --- | --- |
| Highland | – | 0.95 | 0.99 |
| Lowland | 0.05 | – | 0.98 |
| Midland | 0.01 | 0.02 | – |

## Associations of hemoglobin genotypes with morphometric traits

Hemoglobin genotypes had a significant (p<0.05) effect on most measured traits including body weight, chest girth, height at the withers, rump length, and rump height (Table 5). Goats with the Hb$^{BB}$ genotype exhibited higher morphometric traits compared to those with the Hb$^{AA}$ and Hb$^{AB}$ genotypes. In contrast, the Hb$^{AB}$ genotype was associated with a smaller fore canon circumference (p<0.05) than the homozygous types (Table 5). The interaction effects between hemoglobin genotype and agroecological zones was not significant (p>0.05) for most morphometric traits except chest depth (S1 Table). In midland agroecological zone, goats with the Hb$^{AB}$ genotype had greater chest depth than other genotypes. However, in highland and lowland agroecological zones, Hb$^{AB}$ genotypes showed intermediate chest depth values. The smaller chest depth was observed in goats with Hb$^{AA}$ genotypes reared in the lowland and those with Hb$^{BB}$ in the highland agroecological zones (S1 Table). The detailed information on the effect of hemoglobin genotypes on morphometric traits is available in S4 File of the Supporting Information.

**Table 5. Associations of morphometric traits with hemoglobin genotypes in a goat population reared under three agroecological zones.**

| Morphometric traits | Hemoglobin genotypes | | | SEM | P–values |
| --- | --- | --- | --- | --- | --- |
| | AA | AB | BB | | |
| Body weight | 27.5[b] | 27.9[b] | 33.6[a] | 0.545 | <.001 |
| Chest girth | 72.1[b] | 71.9[b] | 76.4[a] | 0.696 | <.001 |
| Height at Withers | 65.9[b] | 66.5[b] | 69.2[a] | 0.502 | <.001 |
| Body length | 65.1 | 64.3 | 65.3 | 0.513 | 0.191 |
| Rump length | 20.3[b] | 20.3[b] | 21.9[a] | 0.206 | <.001 |
| Rump width | 14.6 | 14.6 | 14.7 | 0.201 | 0.880 |
| Rump height | 65.8[b] | 66.1[b] | 68.8[a] | 0.476 | <.001 |
| Chest depth | 30.1 | 30.5 | 30.5 | 0.261 | 0.397 |
| Fore canon circumference | 8.0[a] | 7.9[b] | 8.2[a] | 0.081 | 0.044 |
| Ear length | 13.7 | 13.9 | 14.0 | 0.142 | 0.147 |

a, b =Means between hemoglobin genotypes with different superscript letters are significantly different at p<0.05; SEM = standard error of the mean

## Discussion

The observed variations in body size among the goat populations in the three agroecological zones might be attributed to differences in their adaptation to diverse production environments. Several factors could contribute to this variability including, management practices [49], nutritional variation [50], and selection criteria and intensity [51] which may not be equally favored under different agroecological zones. The selection criteria practiced by farmers in different agroecological zones was varied and depends on the knowledge of livestock owner and production systems [52]. Goat farmers have their own established criteria considered to select their breeding animals in different production systems [42], suggests that genetic improvement programs need to consider production systems and environments. The superiority of goats reared in lowland agroecological zone in the present study was consistent with the findings of [53] and [54], who reported enhanced growth and body measurements in lowland goat populations. In contrast, the lower performances of goats reared in highland agroecological zone may be attributed to the limited availability of browsing pastures, shrubs and bushes which is the main feature of this agroecological zone [54]. It has been also reported that shortage of grazing pasture, goat mobility in crowded communities and disease such as liver fluke, internal parasite and pneumonia were the major constraints in highland agroecological zone [41].

Blood and its components are crucial for assessing and characterizing animal populations [12,13]. Among these, biochemical markers are important to evaluate the genetic variation and similarity of livestock species, even those with the same strain, breeds and closely related lines [55]. Because of their accessibility and biological importance, hemoglobin polymorphisms have been extensively used to study genetic variation in vertebrate animals [56]. The existence of three hemoglobin genotypes (Hb$^{AA}$, Hb$^{AB}$ and Hb$^{BB}$) in the current study was in line with [14,23], who reported similar findings. On the other hand, [27] and [24] reported the occurrence of two genotypes, Hb$^{AA}$ and Hb$^{AB,}$ in West African dwarf sheep and goats, respectively. In contrast, four hemoglobin genotypes (Hb$^{AA}$, Hb$^{AB}$, Hb$^{BB}$ and Hb$^{AC}$) were reported by [57] and [13] for Nigerian goats, which are determined by three codominant alleles, namely, Hb$^{A}$, Hb$^{B}$ and Hb$^{C}$. However, the hemoglobin Hb$^{C}$ allele is rarely observed in animals, and there are different hypotheses regarding its existence. According to [34], the presence of the Hb$^{C}$ allele in goats may be linked to anemia caused by illnesses or environmental stress. Conversely, other researchers have indicated that hemoglobin Hb$^{C}$ may be associated with fetal hemoglobin, which predominantly exists in young animals and diminishes rapidly as the animal reaches maturity [13,36].

The predominance of Hb$^{AB}$ genotype for midland and lowland goats in the present study was in agreement with those reported by [13] for goats in Abuja, [20] for indigenous sheep in Nigeria and [12] for West African dwarf goats. In contrast, [58] reported much lower frequencies of Hb$^{AB}$ in Arbi and Serti goat populations of Tunisia. The higher proportions of Hb$^{AB}$ might be due to the suitability of this genotype under specific environmental conditions because of the natural selection process toward the fixation of this hemoglobin type [59]. Furthermore, goats possessing Hb$^{AB}$ type have demonstrated greater productivity in both weight gain and milk production and suggest its productive advantage under specific environmental conditions [23]. The differences in hemoglobin types among goat populations may be attributed to the variability in altitude and topography, which offers selective advantages in specific geographical regions [20,60]. Several studies have demonstrated associations between hemoglobin variants and environmental adaptability [14,23], suggesting that certain genotypes may be naturally favored in particular agroecological zone [20]. For instance, the Hb$^{A}$ allele has been linked with genetic resistance to helminth infection [27] and essential for survival in low-oxygen conditions of highland area [61,62]. In addition to adaptability, hemoglobin types have also been associated with various productivity traits across livestock species. These include meat quality [63], weaning weight [64], milk and milk fat yield [33,65], and productivity [66] as well as body weight and scrotal size of different livestock species [34].

The higher frequency of Hb$^{BB}$ genotype and Hb$^{B}$ allele in the lowland goat population could be associated with a decreased hematocrit value, lower blood viscosity, and higher water content of Hb$^{B}$ [1,6] which allows animals to adapt to the hot tropical environments. Consistent with the present findings, [23] have reported a high prevalence of Hb$^{B}$ in animals reared in lowland regions. These variations in hemoglobin Hb$^{A}$ and Hb$^{B}$ levels is considered an opportunity to design breeding programs in different geographical regions [6,62].

The predominance of HbAA genotypes in highland goats could be attributed to its adaptation to high-altitude, which is characterized by its cold temperature in specific months of the year [14,23]. The greater abundance of the HbA allele could be linked to its higher oxygen affinity, which confers a survival advantage under hypoxic environmental conditions [13,56,67]. This observation aligns with [68] and [69] who reported a high frequency of HbAA in goats. Likewise, the abundance of allele HbA has been reported for sheep and goats [14], red sokoto and WAD goats [7], Libyan goats [30] and WAD goats in Nigeria [23,24]. The genotype frequency exhibited similar trend among female and male goats, suggesting hemoglobin polymorphisms does not influenced by sexual dimorphisms [59]. However, owing to the imbalance in sample size between males and females, we are unable to conclude the effects of sex on hemoglobin genotypes.

The highland and lowland goat populations are under HWE suggests, these populations have undergone random mating with minimal or no artificial selection pressure. Under HWE, genotype frequencies remain constant across generations in the absence of evolutionary forces, implying a stable population structure [70]. Similar observations have been reported by several authors for indigenous sheep, goat, and cattle populations raised in different parts of Africa [20,36,39,56]. On the other hand, the deviation of midland goats from HWE may be attributed to the management system, mating pattern, and population size [24]. Supporting evidence of such deviations has been reported by several studies including [71] for the Brazilian horse, [20] for Nigerian sheep, [34] for red Sokoto goats, [37] for WAD sheep, [55] for Nigerian turkey and [72] for Ethiopian sheep populations. In smallholder systems, breeding practices including, selection for preferred traits, animal exchange for cultural ceremonies and mating purposes as well as inbreeding due to repeated use of a few selected breeding animals are the causes of deviation from HWE [73,74]. Deviation from the HWE may also arise from non-random mating, inbreeding, natural selection and genetic drift, in small populations, which alter gene and genotype frequency [75].

Genetic diversity within individual populations reduces the risk of recessive gene-associated diseases, and increases the chance of survival in a wide range of environments [76].The heterozygosity values in this study were higher than the reported for Nigerian goat populations [7,8], two Tunisian goats [58], and three turkey populations [11]. The heterozygosity values in the present study are within the recommended range of 0.3 and 0.8 for genetic markers to be useful for measuring genetic diversity [77]. The level of heterozygosity in the present study indicates the existence of moderate genetic variability between the goat populations, which ensures the potential of the goat population to adapt to the changing production environment and breeding goals [25]. Heterozygosity is a valuable measure of genetic variability, with low values indicating reduced genetic variability [16]. Heterozygous individuals are generally more resilient to natural selection and exhibit lower levels of inbreeding than homozygotes [34]. Populations with higher intra-breed similarity and fewer polymorphic loci tend to have lower heterozygosity levels, thereby limiting their genetic variability [21].

The results of Nei's genetic identity indicate that the goat populations reared in highland agroecological zone are more similar to those reared in midland agroecological zone (0.99). The Nei's genetic identity of the highland goat population in the present study was greater than that reported by [7]. The standard genetic distance (D) between the goat populations indicated the existence of genetic differentiation between the populations in the three agroecological zones. The greatest distance between highland and lowland goats might be associated with the geographical differences between these goat populations, in which the chance of population admixture due to the marketing and other sociocultural interactions is very unlikely.

The significant influence of hemoglobin genotypes on morphometric traits were in consonance with [29] who reported that different morphometric traits were influenced by hemoglobin variants in red Sokoto goats. Similarly, [35] reported that neck length, neck width, and ear length were affected by hemoglobin type in Nigerian sheep populations. The highest performances of goats with HbBB genotypes were in consistent with [29] who reported that goats with HbBB type demonstrated better body weight and morphometric characteristics. The authors further claimed that the selection of goats with HbBB types might result in larger-framed animals with higher meat yield and market value. In contrast, [24] reported that WAD goats with the HbAA genotype had higher body weight and chest girths while [34] found that sheep with HbAB genotypes had greater body weights. Similarly, [27] noted that sheep with HbAB genotype exhibited long hair and horns.

The largest chest depth of goats with Hb$^{AB}$ genotypes in midland agroecological zone may be due to the suitability of this specific genotype to this environments. Different polymorphic hemoglobin alleles may be linked to traits of economic importance due to pleiotropic effect or general heterozygosity and have a vital role in the genetic improvements of farm animals [25]. Hemoglobin consists of two α-globin and two β-globin subunits, which are encoded by separate genes whose interactions determine oxygen-binding characters. The α-globin subunits are encoded by two genes, HbA1 and HbA2, which are differentially expressed and function in oxygen transport from the lungs to the tissues, facilitating oxidative metabolism [78,79]. Variation in hemoglobin variants can therefore influence growth and physical traits, given hemoglobin's critical role in oxygen delivery [59]. For instance, in high-altitude areas, certain hemoglobin variants improve oxygen transport and support higher growth rates and larger body size in hypoxic production environments. The superior performance of goats with Hb$^{BB}$ genotype in the present study may be attributed to the role of this genotype in promoting growth and development.

## Conclusion

The current study indicates that the highland goat population exhibited predominant frequency of Hb$^{AA}$ genotype, while goats in both lowland and midland agroecological zones have shown higher proportions of the Hb$^{AB}$ genotype. The results of the chi-square test revealed a significant deviation from Hardy-Weinberg equilibrium in goats reared under midland conditions, which suggests the presence of potential selection pressures and admixture due to migration. The observed associations of hemoglobin variants with morphometric characteristics may suggest that hemoglobin polymorphisms could be used as reliable biomarkers for the genetic improvement of indigenous goat population. Future studies are recommended by incorporating larger sample sizes, whole-genome approaches, and environmental variables to provide a more comprehensive understanding of the genetic diversity and adaptive potential of indigenous goats.

## Supporting information

**S1 Table.  The interaction effects of hemoglobin genotype and agroecological zone.**
(DOCX)

**S1 File.  Raw data used for morphometric traits and hemoglobin genotype analysis.**
(XLSX)

**S2 File.  Pop gene results of hemoglobin for the effect of agroecological zone.**
(TXT)

**S3 File.  Pop gene results of hemoglobin for the effect of sex.**
(TXT)

**S4 File.  The GLM SAS results for the effect of hemoglobin on morphometric traits.**
(MHT)

## Acknowledgments

We are thankful to households and development agents who assist during data collection.

## Author contributions

**Conceptualization:** Kebede Tilahun, Aberra Melesse, Simret Betsha.

**Data curation:** Kebede Tilahun.

**Formal analysis:** Kebede Tilahun, Aberra Melesse.

**Funding acquisition:** Aberra Melesse, Simret Betsha.

**Investigation:** Kebede Tilahun.

**Methodology:** Kebede Tilahun, Aberra Melesse, Simret Betsha.

**Project administration:** Simret Betsha.

**Resources:** Kebede Tilahun, Aberra Melesse, Simret Betsha.

**Software:** Kebede Tilahun, Aberra Melesse, Simret Betsha.

**Supervision:** Aberra Melesse, Simret Betsha.

**Validation:** Kebede Tilahun, Aberra Melesse, Simret Betsha.

**Visualization:** Kebede Tilahun, Aberra Melesse, Simret Betsha.

**Writing – original draft:** Kebede Tilahun.

**Writing – review & editing:** Kebede Tilahun, Aberra Melesse, Simret Betsha.

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
