## [Decision Letter · Decision Letter 0]

3 Mar 2025

PONE-D-24-50965Assessing the genetic diversity of Ethiopian indigenous goat ecotypes at the hemoglobin locus and its associations with morphometric traitsPLOS ONE

Dear Dr. Tilahun,

Thank you for submitting your manuscript to PLOS ONE. After careful consideration, we feel that it has merit but does not fully meet PLOS ONE’s publication criteria as it currently stands. Therefore, we invite you to submit a revised version of the manuscript that addresses the points raised during the review process.

We look forward to receiving your revised manuscript.

Kind regards,

Muhammad Abdul Rehman Rashid, PhD

Academic Editor

PLOS ONE

Journal Requirements:

3. In your Methods section, please provide additional details regarding participant consent from the owners of the animals. In the ethics statement in the Methods and online submission information, please ensure that you have specified (1) whether consent was informed and (2) what type you obtained (for instance, written or verbal). If the need for consent was waived by the ethics committee, please include this information.

5. In the online submission form, you indicated that the data will be made available upon reasonable request. 

Reviewers' comments:

Reviewer's Responses to Questions

**Comments to the Author**

1. Is the manuscript technically sound, and do the data support the conclusions?

Reviewer #1: Yes

Reviewer #2: Yes

2. Has the statistical analysis been performed appropriately and rigorously? 

Reviewer #1: Yes

Reviewer #2: Yes

3. Have the authors made all data underlying the findings in their manuscript fully available?

Reviewer #1: Yes

Reviewer #2: Yes

4. Is the manuscript presented in an intelligible fashion and written in standard English?

Reviewer #1: Yes

Reviewer #2: Yes

5. Review Comments to the Author

Reviewer #1: The manuscript presents valuable findings on hemoglobin polymorphisms and their relationship with morphometric traits in Ethiopian goats. The manuscript has the potential to make a significant contribution to livestock genetics and breeding strategies by applying these revisions

1. The study aims to examine genetic diversity and its relationship with morphometric traits. However, the introduction lacks a clear articulation of how this knowledge could be practically applied. Revise the introduction to explicitly connect the findings to broader goals, such as improving productivity or resilience in goat populations.

2. Clarify and simplify the abstract for improved readability. For example, "The Hb distribution varied between agro-ecologies and the highest (0.47) HbAA genotype was observed in highland goats while the highest HbAB were observed in the lowland and midland goat populations..." could be condensed to "HbAA was most frequent in highland goats (47%), while HbAB dominated in lowland and midland populations."

3. Revise the introduction to highlight the study’s practical relevance in breeding programs and livestock productivity.

4. While ethical approvals are mentioned, the manuscript does not address the practical challenges of implementing genotype-based selection in smallholder farming systems, where resources may be limited. Including a paragraph on potential implementation challenges and solutions would strengthen the study.

5. Provide additional context on deviations from HWE and heterozygosity results compared to global benchmarks.

6. Strengthen language and grammar throughout the manuscript.

7. write name of equation in line 180 and method in line 181.

8. Include visual aids (graphs, charts) to summarize key results.

Reviewer #2: Dear Authors,

The study entitled “Assessing the genetic diversity of Ethiopian indigenous goat ecotypes at the hemoglobin locus and its associations with morphometric traits” examines the genetic diversity of Ethiopian indigenous goat ecotypes at the hemoglobin (Hb) locus and its association with morphometric traits. Blood samples from 225 mature goats of both sexes were analyzed using gel electrophoresis, revealing three hemoglobin genotypes (HbAA, HbAB, and HbBB) controlled by two co-dominant alleles (HbA and HbB), with frequencies of 0.57 and 0.43, respectively. The distribution of Hb genotypes varied across agro-ecological zones, with HbAA being most common in highland goats (0.47), HbAB in midland goats (0.61), and HbBB in lowland goats (0.29). A significant deviation (P<0.05) from Hardy-Weinberg Equilibrium (HWE) was observed in midland goats, while highland and lowland populations remained in equilibrium. The study also found genetic variability (heterozygosity 0.360–0.613), indicating potential for genetic improvement. Additionally, Hb polymorphisms were significantly associated (P<0.05) with several morphometric traits, with HbBB homozygous goats exhibiting superior body weight, heart girth, and rump measurements. These findings suggest that Hb polymorphisms could serve as genetic markers for selective breeding programs to enhance productivity and adaptability in Ethiopian indigenous goats.

The study has several limitations that should be considered when interpreting the results. First, the sample size (225 goats) may not be sufficiently large to capture the full extent of genetic variability within Ethiopian indigenous goat populations, potentially limiting the generalizability of the findings. Second, while the study identifies associations between hemoglobin polymorphisms and morphometric traits, it does not establish causal relationships, and other genetic or environmental factors influencing these traits were not explored. Third, the study focuses solely on the hemoglobin locus, whereas multiple genes contribute to morphometric traits and adaptability, limiting a comprehensive genetic assessment. Fourth, although significant differences were observed between agro-ecological zones, the study does not account for management practices, nutritional status, or selective breeding histories, which could influence the results. Fifth, functional validation through molecular or physiological studies was not conducted to confirm the potential role of Hb polymorphisms in adaptation or productivity. Finally, the deviation from Hardy-Weinberg Equilibrium (HWE) in midland goats suggests possible selection, mutation, migration, or non-random mating, but these factors were not further investigated. Future studies should incorporate larger sample sizes, whole-genome approaches, and environmental variables to provide a more comprehensive understanding of the genetic diversity and adaptive potential of indigenous goats.

Apart from these technical evaluations, another important point concerns the capacity of the data. The study may be highly valuable at the local level. However, its contribution to goat breeding or goat genetics on an international scale is quite limited. Local breeds are crucial, and their genetic characterization provides valuable data for biodiversity conservation. However, the international applicability of such datasets and their adaptation to larger populations are quite challenging. I believe it would be more appropriate to publish these findings in regional or specialized journals that focus on local livestock genetics and biodiversity.

Best regards.

6. PLOS authors have the option to publish the peer review history of their article (what does this mean? ). If published, this will include your full peer review and any attached files.

**Do you want your identity to be public for this peer review?** For information about this choice, including consent withdrawal, please see our Privacy Policy .

Reviewer #1: No

Reviewer #2: **Yes: ** Sena Ardicli

---

## [Author Response · Author response to Decision Letter 1]

12 May 2025

Dear reviewers and editor, Thank you for the feedback and suggestions forwarded for the improvement of the manuscript.

Response to reviewer #1

Dear reviewer, Thank you very much.

1. The introduction was revised based on the provided comment. It now elaborates on genetic diversity, emphasizing its essential role in driving genetic improvement and facilitating adaptation. Moreover, it highlights the importance of Hb polymorphisms in examining genetic diversity, adaptation, and productivity and underscores their potential as markers for selecting animals in genetic improvement programs.

2. The abstract has been improved to make it easier to understand, especially in lines 31-34, 36-37, and 39-42.

3.The role of Hb polymorphisms in livestock breeding programs and genetic improvement was highlighted in the introduction, especially from lines 62-83.

4. The challenges of genotype-based selection under a smallholder farming system and potential solutions were presented in detail (lines 437-462)

5. The deviation from HWE and its context from different perspectives, including relevant scientific assumptions, with comparisons to global benchmarks, are provided in lines 364-383.

The language and grammar throughout the manuscript have been thoroughly revised and improved. The names of equations and methods are now provided in lines 179 and 180.

6. The distribution pattern of total Hb is presented in a pie chart and provided as supplementary material (S1 Fig.)

The distribution of Hb genotypes based on agroecology is shown in a chart (S2 Fig.). (S1 and S2 Figs) were provided as supplementary files. However, further summarization of results through visual aids was not feasible. For example, variables such as heterozygosity, genetic distance, and association of morphometric traits with genotype data are not suitable to display in graphical form.

Response to reviewer #2

Dear reviewer, Thank you very much

1. The sample size of 225 goats may appear limited when considering the total population of Ethiopian goats, estimated at over 46 million heads and comprising more than 14 distinct breeds. Our study focused on southeastern Ethiopia. We purposefully selected goats from three representative districts across three agroecological zones: highland, midland, and lowland. This approach ensured the inclusion of variations in management practices, nutritional status, and production systems. Given this targeted design, we believe the sample is appropriate for assessing genetic diversity at the Hb locus and its association with key performance traits within the study area. Moreover, many peer-reviewed studies published in international journals have utilized comparable or smaller sample sizes when examining genetic markers in livestock. To support the adequacy of our sample, we have listed some articles with their sample size and DOI that provide examples of such studies. These references demonstrate that our sample size is consistent with accepted scientific practice, particularly in studies focused on local breeds and targeted loci such as hemoglobin variants. Therefore, we are confident that our findings provide meaningful insights, particularly for the regions and contexts studied.

Some examples of published articeles

Gwaza et al., 2019: sample size 40.https://doi.org/10.18488/journal.57.2019.81.38.47(Goat)

Agaviezor et al.,2013 sample size 60. https://doi.org/10.13140/2.1.1552.2246(Goat)

Osaiyuwu & Salako, 2018 sample size 100. https://doi.org/10.5897/ajb2015.14655(sheep)

Važić et al., 2017 sample size 189. https://doi.org/10.2298/GENSR1701151V(Sheep)

Orunmuyi et al., 2020 sample size 155. https://doi.org/10.1007/s11250-020-02257-y(Poultry)

Chebo et al., 2024 sample size 284. https://doi.org/10.1007/s11250-024-04086-9(Poultry)

Pal and Mummed, 2014 sample size 105. https://doi.org/10.14202/vetworld.2014.229-233(Cattle)

2. While our study focuses on association rather than causation, there are several biological reasons to explore the link between the Hb variant and morphometric traits and other traits related to production, disease resistance, and adaptability. Additionally, there was budget limitation to explore other genetics and environmental factors. Hemoglobin is a complex protein composed of two α-globin and two β-globin subunits, which are encoded by separate genes. These subunits influence the oxygen-biding capacity and transport efficiency, which are critical for cellular metabolism and overall physiological growth. Variations in Hb structure can lead to differences in oxygen delivery to tissues, potentially affecting growth rate, body size, and other morphometric traits, especially under varying environmental conditions. Some Hb variants also have a selective advantage against disease and increase the survivability of animals. Additional context and explanation regarding these mechanisms have been provided in the discussion part (lines 317- 328).

3. We acknowledge that production and morphometric traits are polygenic, influenced by multiple genes with varying effects (e.g., QTLs). However, the Hb locus is known to be highly polymorphic across species and may play a significant role in productivity and adaptability. These effects could arise through mechanisms such as dominance, pleiotropic and epistasis, and interactions with other locus and general heterozygosity. Due to its polymorphic nature and physiological importance, we believe the Hb locus contributes meaningfully to the expression of performance and adaptive traits.

# Additionally, we recommended the integration of broader genomic tools such as genetic mapping and microarray-based technologies to further examine gene expression and identify genes affecting a wider range of traits.

The role of hemoglobin in adaptability and performance traits is discussed in the manuscript (lines 66-77 and 420-435).

4.Since the study was carried out under on-farm conditions, agro ecological variation inherently includes differences in nutrition, management practices, feed and water availability, and overall husbandry systems. These factors are integral components of agrological variation and are important for evaluating the actual genetic potential, adaptability, and performance of animals in their natural production environments. By considering these real-world variations, our study aims to ensure that future genetic improvement programs are aligned with the practical conditions in which the animals are raised.

To further strengthen the result we have provided detailed information on:

1. The management of experimental animals in the methodology section (lines 122-131);

2. The possible reasons for performance differences among the three agroecologies, from the local study context and broader global insights, are in the discussion section (lines 270-277 and 280-286).

The interaction between Hb genotype and agroecology, with the results presented in the results section (lines 255-260) and summarized in Supplementary Table 1 (S1 Table).

5. Thank you for raising this important point. Our study did not include functional validation through molecular or physiological experiments to directly confirm the role of Hb polymorphisms in adaptation or productivity. This is due to lack of budget to undertake other molecular and physiological study. The allocated budget is only for hemoglobin polymorphisms study (for blood sample collection and analysis). However, our research was designed as an exploratory genetic and phenotypic association study under field conditions, aiming to identify potential links between Hb variants and traits of interest in diverse agroecologies. The functional significance of hemoglobin polymorphisms has been widely discussed in the scientific literature, and we have incorporated relevant findings to support our interpretations. These are articulated in the introduction (from lines 66-74) and further discussed in the discussion section (lines 319-325).

6. Thank you for this valuable comment. Indeed, deviations from Hardy-Weinberg Equilibrium (HWE) are common in natural populations, particularly under smallholder farming systems where ideal HWE assumptions, such as random mating, no selection, mutation, migration, or genetic drift are rarely met. In our study, we discussed the possible explanations for the observed deviation in midland goats, including natural and artificial selection, gene flow through migration (for feed or market purposes), and small effective population sizes. These points are elaborated in the manuscript (lines 370-383) within local and broader contexts.

While we acknowledge that the current study did not investigate each of these evolutionary forces in detail due to its scope and design, we agree with the reviewer that future research should employ larger sample sizes, incorporate whole-genome approaches, and consider environmental and management variables to explore the genetic architecture and adaptive potential of indigenous goat populations. These recommendations have been incorporated in the conclusion section.

7.We fully acknowledge the importance of aligning scientific contributions with local relevance and broader international applicability. Hemoglobin polymorphisms are of global scientific interest due to their well-documented polymorphic nature, simple pattern of inheritance, and their association with key performance and adaptive traits including, productivity, disease resistance, adaptability, and fertility. As demonstrated in our study and the literature, Hb variants have functional relevance and are valuable genetic markers in selective breeding programs. In addition, Hb genotyping is low-cost, accessible, and easy to interpret, making it a practical tool for many developing countries where advanced genomic technologies may not be widely available. We believe this study provides insights that go beyond local application. Although focused on southeastern Ethiopia, the methodology, findings, and implications regarding Hb variation and its phenotypic associations are relevant for researchers, breeders, and policymakers working with indigenous livestock populations under low-input systems globally. Most importantly, the integration of Hb genotypes with morphometric traits enhances the broader significance of the work by linking molecular markers with observable performance characteristics. We appreciate the suggestion regarding publication venue, but respectfully submit that our study addresses pressing questions in livestock genetics and adaptation that are also of international relevance, particularly in the context of biodiversity conservation and climate-resilient breeding strategies.

Response to academic editor

Dear editor thank you very much

1. Thank you for the reminder. The manuscript has been formatted according to the Plos One Author guidelines, including file naming conventions.

2. The corresponding author’s existing ORCID iD has been validated in Editorial Manager.

3. Thank you for the suggestion. Participant consent has been included in the methods section (Lines 133-139). Initially, some farmers were hesitant expressing concerns that blood sample collection might harm their animals by causing weight loss or exposing them to diseases. To address this the researchers reassured the farmers that all blood samples would be collected by trained professionals with veterinary expertise and that the procedure would not harm the animals. An informal oral agreement was reached with each participating farmer. Following this mutual understanding, the blood and morphometric data were collected.

The ethical standard of the research was reviewed and approved by the Research Ethics Review Committee (RERC) of Hawassa University. This information is provided in the methodology section under Ethics approval (Lines 95-97).

4. Thank you for pointing this out. We have now provided the correct grant number and included the names of the award recipients in the “Funding Information” section.

5. Thank you for the clarification. All relevant data including raw data, SAS and pop gene outputs were added as supplementary information at the end of the manuscript after the reference list

---

## [Decision Letter · Decision Letter 1]

25 Jun 2025

PONE-D-24-50965R1Assessing the genetic diversity of Ethiopian indigenous goat ecotypes at the hemoglobin locus and its associations with morphometric traitsPLOS ONE

Dear Dr. Tilahun,

Thank you for submitting your manuscript to PLOS ONE. After careful consideration, we feel that it has merit but does not fully meet PLOS ONE’s publication criteria as it currently stands. Therefore, we invite you to submit a revised version of the manuscript that addresses the points raised during the review process.

We look forward to receiving your revised manuscript.

Kind regards,

Muhammad Abdul Rehman Rashid, PhD

Academic Editor

PLOS ONE

Journal Requirements:

Reviewers' comments:

Reviewer's Responses to Questions

**Comments to the Author**

1. If the authors have adequately addressed your comments raised in a previous round of review and you feel that this manuscript is now acceptable for publication, you may indicate that here to bypass the “Comments to the Author” section, enter your conflict of interest statement in the “Confidential to Editor” section, and submit your "Accept" recommendation.

Reviewer #3: (No Response)

Reviewer #4: All comments have been addressed

Reviewer #5: All comments have been addressed

2. Is the manuscript technically sound, and do the data support the conclusions?

Reviewer #3: No

Reviewer #4: Yes

Reviewer #5: Yes

3. Has the statistical analysis been performed appropriately and rigorously? 

Reviewer #3: No

Reviewer #4: Yes

Reviewer #5: I Don't Know

4. Have the authors made all data underlying the findings in their manuscript fully available?

Reviewer #3: Yes

Reviewer #4: Yes

Reviewer #5: No

5. Is the manuscript presented in an intelligible fashion and written in standard English?

Reviewer #3: Yes

Reviewer #4: Yes

Reviewer #5: Yes

6. Review Comments to the Author

Reviewer #3: Although the study contributes to the knowledge in its field, the following points should be taken into consideration and necessary arrangements and corrections should be made.

Introduction:

1) Lines 120-124: Here it is stated that there are no studies on Hb in Ethiopia. However, a brief search revealed the following two studies. Therefore, these studies should be evaluated in this section and necessary citations should be made.

Simachew A., Bekele E., 2013. Genetic variatıon of some goat populations in Ethiopia by means of blood protein polymorphism. Ethiop. J. Biol. Sci. 12(2): 169-186,

Simachew, Addis, 2002. Characterization of some Goat Populations in Ethiopia by means ofBlood Protein Polymorphism. Thessis. https://hdl.handle.net/10568/67965

Materials and Methods

2) No information was given on the age of the goats used in this study. Were goats of the same age or different ages used as experimental animals? Considering that morphometric characteristics were also determined in the study, ages are important. Because morphometric characteristics will be affected by age.

3) Additional information on the morphometric characteristics to be obtained from the experimental animals is required. In addition, relevant references should be cited to explain the methodology by which these characteristics will be determined.

4) It is stated that POPGENE32 program was used in the data analysis section. First, this program should be cited as indicated below. Secondly, all of the calculations mentioned in lines 176-191 are made with the help of this program. Therefore, there is no need to write equations or give details about the calculations.

YEH, FRANCIS C., YANG, R-C., BOYLE, TIMOTHY, B.J., YE, Z-H., and MAO, JUDY X. 1997. POPGENE, The user-friendly shareware for population genetic analysis. Molecular Biology and Biotechnology Centre, University of Alberta, Canada.

or

YEH, F.C. and BOYLE, T.J.B. 1997. Population genetic analysis of co-dominant and dominant markers and quantitative traits. Belgian Journal of Botany 129: 157.

Results

5) In Table 2, the number of genotypes considering both sexes and the number of genotypes in the overall total are incompatible. For example, the HbAA genotype is stated as 76 in the overall total. In the classification by sex, the HbAA genotype was reported as 35 and 12 in females and males, respectively. The total number of HbAA genotypes in these two sexes is 47. The case is the same for other genotypes.

Reviewer #4: Title: Assessing the genetic diversity of Ethiopian indigenous goat ecotypes at the hemoglobin locus and its associations with morphometric traits

General comment,

The manuscript presents a valuable concept by focusing on the understanding and documentation of genetic diversity in goats using protein markers. The authors have effectively addressed the concerns raised by the previous reviewers, resulting in a clearer and more readable manuscript. However, the manuscript will benefit if the authors consider the following minor comments.

Line 28: Relevant morphometric traits …. Please state the traits

Line 29: The red cells were subjected … The red blood cells were subjected…

Line 39 - : In conclusion, the Hb polymorphisms could serve as a genetic marker for selective breeding programs to enhance productivity and adaptability of indigenous goats.

- Do you have any evidence that which ones are adaptive/productive and which ones are not? If yes, please state it in one sentence

Line 41: A genome-wide association study is recommended to validate the association of Hb variants with economically important traits.

- This is wrong and should be deleted

Line 69: … exhibit greater tolerance under unfavorable conditions than … what unfavourable condition – state the condition

Line 117: … digital portable weighing balance having a 50kg capacity. State the precision of the weighing scale

Line 146: Hemoglobin electrophoresis

- State how blood was collected, transported and stored before going to lab analysis.

Line 185 … The statistical model used to test the relationships between hemoglobin genotypes and morphometric variables was as follows:

Yij = μ+Ai +Hj+ e

- This is a GLM model to check the effect of HB type on performance, not a correlation model – you need to have this correlation model if you want the association

Line 210, S1 and S2. This is an important information and is the main result - need to be presented in the main document.

Line 437: Challenges of genotype-based selection under smallholder farming

- I do not understand the importance of this section – please remove it.

Reviewer #5: The authors need to provide the gel picture of electrophoresis and incorporate it in the manuscript.

7. PLOS authors have the option to publish the peer review history of their article (what does this mean? ). If published, this will include your full peer review and any attached files.

**Do you want your identity to be public for this peer review?** For information about this choice, including consent withdrawal, please see our Privacy Policy .

Reviewer #3: No

Reviewer #4: **Yes: ** Mengistie Taye

Reviewer #5: **Yes: ** Ishraq Hussain

---

## [Author Response · Author response to Decision Letter 2]

30 Jul 2025

RESPONSE TO REVIEWER #3

Reviewer comment:

I. Introduction:

1. Reviewer comments: Lines 120-124: Here it is stated that there are no studies on Hb in Ethiopia. However, a brief search revealed the following two studies. Therefore, these studies should be evaluated in this section, and necessary citations should be made. Simachew A., Bekele E., 2013. Genetic variatıon of some goat populations in Ethiopia by means of blood protein polymorphism. Ethiop. J. Biol. Sci. 12(2): 169-186, Simachew, Addis, 2002. Characterization of some Goat Populations in Ethiopia by means of Blood Protein Polymorphism. Thessis. https://hdl.handle.net/10568/67965

Author's response: Thank you for the suggestions. We have evaluated and cited these articles appropriately. After careful examination of these two works, they address the same topic, similar sample size, study area, methods, and results. This is because the article published is the version of the thesis, and therefore, we have cited the recent one (2013). We have highlighted this study in the introduction section (Lines 88-95).

II. Materials and methods

1. Reviewer comment: No information was given on the age of the goats used in this study. Were goats of the same age or different ages used as experimental animals? Considering that morphometric characteristics were also determined in the study, ages are important. Because morphometric characteristics will be affected by age.

Author's responses: We have sampled adult goats with similar age groups (4PPI), i.e., four years and above. The age was estimated from their dentition as proposed by Ebert and Solaiman (2010). This information has been stated (lines 120-122 and 127-129).

2. Reviewer comment: Additional information on the morphometric characteristics to be obtained from the experimental animals is required. In addition, relevant references should be cited to explain the methodology by which these characteristics will be determined.

Author's response: We have collected some morphometric traits from the experimental goats to evaluate the effect of Hb genotype on those traits. The morphometric traits were measured by following the FAO guidelines for phenotypic description of animals (FAO, 2012).

3. Reviewer comment: It is stated that the POPGENE32 program was used in the data analysis section. First, this program should be cited as indicated below. Secondly, all of the calculations mentioned in lines 176-191 are made with the help of this program. Therefore, there is no need to write equations or give details about the calculations.

Author's responses: The POPGENE32 has been properly cited as recommended. Yeh FC, and Boyle TJB. (1997). Population genetic analysis of co-dominant and dominant markers and quantitative traits. Belgian Journal of Botany 129: 157. All the formulas that indicate manual calculation have been deleted lines (177-179).

III. RESULTS

1. Reviewer Comment: In Table 2 the number of genotypes considering both sex and the number of genotypes in the overall total is incompatible.

Author's responses: After reanalyzing the sex data using the pop gene, we verified all of the errors and made the necessary corrections as indicated by the lines (225–227) and in Table 2. Additionally, fig. 1 shows the total genotype frequency. Pop gene output for the effects of sex and agroecological zone is available as supplemental materials files S2 and S3 Txt.

RESPONSE TO REVIEWER #4

Reviewer comment:

I. Abstract:

1. Reviewer comment: Line 28: Relevant morphometric traits …. Please state the traits.

Author's responses: Thank you for the suggestions. The collected morphometric traits have been included (Lines 27-29)

2. Reviewer comment: Line 29: The red cells were subjected … The red blood cells were subjected

Author's responses: Done (Line 29)

3. Reviewer comment: Line 39: In conclusion, the Hb polymorphisms could serve as a genetic marker for selective breeding programs to enhance productivity and adaptability of indigenous goats. Do you have any evidence that which ones are adaptive/productive and which ones are not? If yes, please state it in one sentence

Author's responses: There are different reports regarding the role of Hb in the adaptation and productivity of animals. It has been reported that HbAA is mostly known for its high oxygen affinity and is found in a higher proportion of animals in high-altitude areas and is resistant to helminth infection. At the same time, there are also reports that indicate the role of this genotype in productivity traits. Whereas the HbBB is known for its high water content, low viscosity, and abundance in animals living in lowland arid areas. In addition, there are different reports regarding the role of Hb variants in productivity traits. However, to explicitly state/articulate which one is adaptive and which one is productive, further study may be required. Generally, it’s difficult to generalize and specifically ascertain and articulate the role of each genotype independently based on the existing data. However, we can say generally hemoglobin has a role in adaptation and production. Therefore, we have revised the statement.

4. Reviewer comment: Line 41: A genome-wide association study is recommended to validate the association of Hb variants with economically important traits. - This is wrong and should be deleted

Author's responses: This sentence has been removed as recommended.

II. INTRODUCTION

1. Reviewer comment: Line 69: … exhibit greater tolerance under unfavorable conditions than … what unfavorable condition—state the condition.

Author's responses: The hemoglobin HbAA genotypes have a selective advantage in high-altitude areas due to their high oxygen affinity, higher Hb concentration, and packed cell volume. This feature allows them to live in higher altitudes, unfavorable feed, and resistance to helminth infections. Therefore, the unfavorable conditions include low-quality feed, low temperature (cold area), oxygen-deficient conditions, and helminth infections (line 68).

III. Materials and methods

1. Reviewer comment: Line 117: … digital portable weighing balances having a 50 kg capacity. State the precision of the weighing scale

Author's response: We have been stated (line 127). The digital portable scale with high accuracy and a precision of 10 g was utilized to measure live body weight.

2. Reviewer comment: Line 146: Hemoglobin electrophoresis—State how blood was collected, transported, and stored before going to lab analysis.

Author's response: The blood sample collection procedures and storage and transport system have been presented (Lines 152-158).

3. Reviewer comment: Line 185 … The statistical model:

Yij=μ+Ai+Hj+e

This is a GLM model to check the effect of HB type on performance, not a correlation model—you need to have this correlation model if you want the association

Author's responses: We have used this model to test the effect of hemoglobin genotypes on morphometric traits. Morphometric data were collected from similar animals in which a blood sample was collected. The blood samples and morphometric data were labeled with similar numbers to indicate which linear data was collected from which animals. Then the Hb genotypes were inserted in Excel, which contains morphometric data. For example, goat 1HF is a female animal sampled from highland agroecology. The genotype and morphometric measurement of the 1HF goat are filled directly on the Excel sheet. That is the case we call association. Moreover, the Hb genotypes are categorical (AA, AB, and BB), and their values cannot be expressed in numbers. Thus, while associating them with morphometric traits, it’s difficult to use a correlation model. Instead, we have used the PROC GLM model by considering the Hb genotypes as a fixed effect.

IV. Results

1. Reviewer comment: Line 210, S1 and S2. This is an important information and is the main result - need to be presented in the main document.

Author's response: Both figures have been presented in the main document (Fig 1 and Fig 2).

V. Discussion

1. Reviewer comment: Line 437: Challenges of genotype-based selection under smallholder farming. I do not understand the importance of this section—please remove it.

Author’s responses: We have removed this part.

RESPONSES TO REVIEWER #5

1. Reviewer comments: The authors need to provide the gel picture of electrophoresis and incorporate it in the manuscript

Author's response: The gel pictures have been provided as supplementary material (S1 Fig).

RESPONSE TO EDITOR

1. Editor comment: Please review your reference list to ensure that it is complete and correct. If you have cited papers that have been retracted, please include the rationale for doing so in the manuscript text, or remove these references and replace them with relevant current references. Any changes to the reference list should be mentioned in the rebuttal letter that accompanies your revised manuscript. If you need to cite a retracted article, indicate the article’s retracted status in the References list and also include a citation and full reference for the retraction notice.

Author’s responses: We have reviewed, edited, and corrected all references in the reference list. We have also ensured all references under the citation are properly cited in the reference list. We have checked the retracted paper using the “Retraction Watch Database” by inserting the author’s name, title, journal, and DOI. Based on the results of the Retraction Watch Database, all references used in the manuscript are not retracted, or no reference is found in retraction status. In addition, we have edited the comma, semicolon, and journal names to make them consistent. Moreover, based on the reviewer’s comments, we have included four (4) new references and removed/deleted about fifteen (15) references with the underlined data.

---

## [Editor Report · Decision Letter 2]

1 Aug 2025

Assessing the genetic diversity of Ethiopian indigenous goat ecotypes at the hemoglobin locus and its associations with morphometric traits

PONE-D-24-50965R2

Dear Dr. Tilahun,

We’re pleased to inform you that your manuscript has been judged scientifically suitable for publication and will be formally accepted for publication once it meets all outstanding technical requirements.

Kind regards,

Muhammad Abdul Rehman Rashid, PhD

Academic Editor

PLOS ONE
---

## [Editor Report · Acceptance letter]

PONE-D-24-50965R2

PLOS ONE

Dear Dr. Tilahun,

I'm pleased to inform you that your manuscript has been deemed suitable for publication in PLOS ONE. Congratulations! Your manuscript is now being handed over to our production team.

Kind regards,

on behalf of

Dr. Muhammad Abdul Rehman Rashid

Academic Editor

PLOS ONE